# Down-Regulation of Lipid Metabolism in the Hepatopancreas of Shrimp *Litopenaeus vannamei* upon Light and Heavy Infection of *Enterocytozoon hepatopenaei*: A Comparative Proteomic Study

**DOI:** 10.3390/ijms231911574

**Published:** 2022-09-30

**Authors:** Yujiao Wu, Jie Chen, Guoli Liao, Mengjiao Hu, Qing Zhang, Xianzhi Meng, Tian Li, Mengxian Long, Xiaodong Fan, Qing Yu, Liping Zhang, Guoqing Pan, Zeyang Zhou

**Affiliations:** 1Chongqing Key Laboratory of Microsporidia Infection and Control, Southwest University, Chongqing 400715, China; 2State Key Laboratory of Silkworm Genome Biology, Southwest University, Chongqing 400715, China; 3Pearl River Fisheries Research Institute, Chinese Academy of Fishery Sciences, Guangzhou 510380, China; 4College of Life Science, Chongqing Normal University, Chongqing 400047, China; 5Chongqing Fisheries Technical Extension Center, Chongqing 400014, China

**Keywords:** *Enterocytozoon hepatopenaei*, *Litopenaeus vannamei*, proteomic, lipid metabolism

## Abstract

*Enterocytozoon hepatopenaei* (EHP) is the pathogen of hepatopancreatic microsporidiosis (HPM) in shrimp. The diseased shrimp *Litopenaeus vannamei* exhibits a slow growth syndrome, which causes severe economic losses. Herein, 4D label-free quantitative proteomics was employed to analyze the hepatopancreas of *L. vannamei* with a light (*EHPptp2* < 10^3^ copies/50 ng hpDNA, L group) and heavy (*EHPptp2* > 10^4^ copies/50 ng hpDNA, H group) load of EHP to better understand the pathogenesis of HPM. Exactly 786 (L group) and 1056 (H group) differentially expressed proteins (DEPs) versus the EHP-free (C group) control were mainly clustered to lipid metabolism, amino acid metabolism, and energy production processing. Compared with the L group, the H group exhibited down-regulation significantly in lipid metabolism, especially in the elongation and degradation of fatty acid, biosynthesis of unsaturated fatty acid, metabolism of α-linolenic acid, sphingolipid, and glycerolipid, as well as juvenile hormone (JH) degradation. Expression pattern analysis showed that the degree of infection was positively correlated with metabolic change. About 479 EHP proteins were detected in infected shrimps, including 95 predicted transporters. These findings suggest that EHP infection induced the consumption of storage lipids and the entire down-regulation of lipid metabolism and the coupling energy production, in addition to the hormone metabolism disorder. These were ultimately responsible for the stunted growth.

## 1. Introduction

*Enterocytozoon hepatopenaei* (EHP) is the pathogen that contributed to hepatopancreatic microsporidiosis (HPM) in three farmed *Penaeus* shrimp (i.e., *P. monodon*, *P. vannamei*, and *P. stylirostris*) and one suspected species (*P. japonicus*) [1]. It was officially named in 2009 when its morphology, histopathology, and phylogenetics were described [2]. Shrimps infected by EHP are not mortal but distinctly display a growth-retarded appearance and, sometimes, induced coinfection [3,4,5]. EHP is a member of the microsporidia (~208 genera), which are single-cell eukaryotes (a sister branch to the Fungi) and very frequently encountered obligate intracellular parasites that can infect both animals and some protists [6,7,8]. Almost half of the known microsporidian genera infect aquatic hosts [9]. Genomes of microsporidia vary in size from 2.3 Mb to 24 Mb and lose most of their genes associated with primary metabolites, resulting in a limited capacity for producing adenosine triphosphate (ATP) [10,11,12]. *Enterocytozoonidae* undergo an extreme reduction of metabolic pathways, suggesting that they completely depend on horizontally acquired ATP and metabolites from the host cell [13]. EHP can be cultured in shrimp and, compared with other microsporidia in *Enterocytozoonidae*, it exhibits a distinctive model in host-parasite interactions.

In 2018, the production of farmed shrimp reached >6 million tonnes, valued at over US$38 billion [14]. *Litopenaeus vannamei* is the most important cultured shrimp widely farmed in Central and South America and Asia (particularly in China, Indonesia, Thailand, and Vietnam), accounting for ~80% of total cultured penaeid shrimp production [15]. *L. vannamei* replaced *P. monodon* and became the main cultured shrimp because it was unaffected by monodon slow growth syndrome (MSGS) in Thailand after 2001 [16]. However, the much higher stocking density of *L. vannamei* might seed a providential condition that accelerates the prevalence of EHP, as microsporidia are generally host-population density-dependent, and EHP can be transmitted directly from shrimp to shrimp by cannibalism and cohabitation [1,17,18]. Although EHP infection is not lethal for shrimp, severe growth retardation would lead to an enormous economic loss.

The hepatopancreas (HP) is an important organ for shrimp and performs the function of digestion, nutrition, and storage [19,20]. Hence, healthy HP is significant for the development of shrimp. EHP is highly dependent on the nutrients and energy of the host, as it would consistently import ATP and primary substrates from host cells throughout its entire parasite life cycle. Nevertheless, not all shrimp infected by EHP exhibited growth retardation; the symptomatic individuals generally had a certain degree of infection. We are interested in how EHP manipulates the nutrient metabolism of the host cell to maintain its proliferation and development. In this study, we employed 4D label-free proteome to survey the global interactions between the hepatopancreas of shrimp and EHP and to explore the potential molecular mechanism that may be associated with the slow-growth syndrome, using a light and high pathogen-loaded HP. The results suggest that down-regulation of lipids, especially through fatty acid metabolism and JH degradation unbalances, was positively correlated with the EHP load, which was likely the main metabolic reason for growth retardation.

## 2. Results

### 2.1. Basic Information of Samples

The stunted shrimp manifested minor body length and weight or weighed less under the same size compared to non-infected shrimp of the same origin and culture condition (Figure 1A). Genomic DNA of the hepatopancreas from slow-growth shrimp and healthy shrimp were used as a template to detect the load of EHP, and RT-qPCR was carried out using primers EHPptp2-192F/R targeting the polar-tube protein 2 of EHP. The absolute real-time quantitative PCR (RT-qPCR) results showed that the copies of EHP were within the scope of 100~106 per 50 ng total hpDNA in stunted shrimp, while there was no signal in the healthy group. Samples were classified into three groups: the healthy control (EHP-free, C group), low EHP load (*EHPptp2* < 10^3^ copies/50 ng hpDNA, L group), and high EHP load (*EHPptp2* > 10^4^ copies/50 ng hpDNA, H group) (Figure 1D). All PCR-detected samples were further verified by microscopic examination with tissue homogenate, showing a result of no more than 10 spores per field if the EHP load was less than 10^3^ copies per 50 ng hpDNA, and more than 100 spores per field were seen when the EHP load exceeded 10^4^ copies per 50 ng hpDNA. Each group contained four biological replicates, and each duplication was a sample mixture of four hepatopancreas at the same partition level. About 3589 proteins from *L. vannamei* and 479 from EHP were identified, and 2781 were quantified. According to the selection standard of FC > 1.5 and *p* < 0.05, 542 up-regulated and 514 down-regulated proteins (H group), as well as 439 up-regulated and 347 down-regulated proteins (L group) of host response for EHP infection, were detected compared with the C group. As for the differential response to different EHP capacities, 144 up-regulated and 128 down-regulated proteins were identified in H compared to L (Figure 1B). To further describe the reliability of the data, the PCA method was used to illustrate the significant differences among the groups and good repeatability of samples within one group (Figure 1C).

### 2.2. EHP-Infected Individuals Exhibited a Significant Down-Regulation in Lipid and Amino Acid Metabolism

To understand what functions these differentially expressed proteins (DEPs) may play in *L. vannamei*, GO and Clusters of Orthologous Groups of proteins (COG) were employed to analyze the properties of the biological process, cellular component, molecular function, and protein origin (COG/KOG). GO functional classification showed that all these DEPs in infected individuals were principally involved in the metabolic and cellular process with binding and/or catalytic activity function (Figure 2A–C). Compared to the C group, DEPs in the H group were classified into the metabolic process (409), cellular process (256), binding (470), and catalytic activity (445) (Figure 2A). While in the L group, 322, 198, 358, and 348 proteins were categorized into the metabolic process, cellular process, binding, and catalytic activity, respectively (Figure 2B). There were 80, 52, 122, and 87 DEPs between H and L groups engaged in metabolic process, cellular process, binding, and catalytic activity, respectively (Figure 2C). The COG classification showed more clearly that these differential proteins were mainly involved in the transport and metabolism of lipids, amino acids, and carbohydrates, as well as energy production and conversion (Figure 2D–F). All function-annotated DEPs were then enriched by KEGG for detailed information on pathways (Figure 3). The results showed that EHP infection led to a nearly entire down-regulation of lipid metabolism, including the degradation and elongation of fatty acid, biosynthesis of unsaturated fatty acid, sphingolipid metabolism, alpha-linolenic acid metabolism, glycosphingolipid biosynthesis, and partial short-chain fatty acid metabolism. Peroxisome and oxidative phosphorylation pathways associated with fatty acid oxidation were also down-regulated. Amino acid metabolism pathways were also distinctly down-regulated, including phenylalanine, tryptophan, histidine, lysine, glycine, serine, threonine, valine, leucine, and isoleucine. Compared with the L group, the H group appeared to have deeper regulation in protein processing in the endoplasmic reticulum, fatty acid degradation, alpha-linolenic acid metabolism, sphingolipid metabolism, biosynthesis of unsaturated fatty acid, beta-alanine, and phenylalanine metabolism. What cannot be neglected is the down-regulation of the hormone pathway, which refers to the degradation and regulation of ecdysone (CYP18A1, ecdysteroid regulated-like protein) and degradation of the juvenile hormone (JH) (Appendix A).

### 2.3. The Host Response Varies with the Severity of EHP Infection

Variation analysis of L and H groups illustrated that the down-regulation of host lipid (fatty acid, sphingolipid, and alpha-linolenic acid) metabolism and hormone (JH) pathways were positively related to the severity of EHP infection. In addition, expression-pattern analysis through the Mufzz method was used to analyze the relevance between differential regulation and gradients of EHP infection. As for protein expression level, lipid and amino acid metabolic pathways were mainly classified into cluster 3 and cluster 6 among six clusters, suggesting that the more severe the infection is, the more significant the effect of down-regulation. The transcriptional regulation of the Wnt signaling pathway, immune-related Toll and IMD signaling pathways, as well as ubiquitin-mediated proteolysis process, were up-regulated along with the EHP infection (cluster 2 and cluster 5 of Figure 4). In fatty acid elongation and degradation, the extent of downregulation was accentuated with increased infection, in according with the expression level. KEGG enrichment of differential proteins between H and L groups further reflected the regulation of fatty acid degradation, sphingolipid metabolism, amino acid metabolism, JH metabolism, and glutathione metabolism, and ubiquitin-associated pathways were significantly correlated with the degree of infection (Figure 5). Detailed DEPs showed that the down-regulated peroxisomal acyl-coenzyme A oxidase 1 (ACOX1) and acetyl-CoA acyltransferase mainly caused the down-regulation of fatty acid degradation and alpha-linolenic acid metabolism, and decreased enzymes involved in sphingolipid metabolism suggested a declining generation of sphingomyelin. Fatty acid degradation and their associated peroxisomal oxidation process are important energy production pathways. In contrast, sphingomyelin is an important component of biomembrane. Reduced aldehyde dehydrogenase was involved in the down-regulation of β-alanine, phenylalanine, and tyrosine metabolism. Fewer JHE-like carboxylesterase (JHEC) meant more severe abnormalities of the JH degradation in the H group. Whereas, a fourfold higher level of ubiquitin-conjugating enzyme indicated that a more active ubiquitin-mediated proteolysis occurred in the H group.

### 2.4. Infected Groups Showed a Coincidental Transcription Level Change Compared with those in the Proteome

To further understand the changes reflected by our data, we performed relative quantitation RT-qPCR to investigate the correlation between protein abundance and gene transcription level. The transcripts of several genes involved in significantly regulated pathways were measured, including genes participating in fatty acids metabolism (acetyl-CoA carboxylase, Triacylglycerol lipase, carnitine O-palmitoyltransferase 1, and delta-9 desaturase), carbohydrate metabolism (lactate dehydrogenase, pyruvate dehydrogenase E1 component subunit beta, and pyruvate carboxylase), hormone metabolism (JHE-like carboxylesterase 1 and Ecdysteroid regulated-like protein), amino acid metabolism (phosphoserine aminotransferase and mitochondrial succinate-semialdehyde dehydrogenase) and immunity (pacifastin light chain-like serine proteinase inhibitor, Prophenoloxidase-2, and lectin). Specific primers were designed for target genes (Appendix A). The transcriptional results showed agreement with protein abundance (Figure 6).

### 2.5. Pathogen Proteins That Participate in Pathogenesis were Identified in EHP-Infected Hepatopancreas

In this study, we identified 479 EHP proteins in total, including 86 uncharacterized proteins and 217 proteins identified in the L group (Figure 7A). Subcellular location prediction indicated that a large number of proteins were located in the nucleus (40%), cytoplasmic proteins accounted for 24%, and mitochondrial proteins made up 14%. The rest were exited in the plasma membrane, endoplasmic reticulum, and so on (Figure 7B). Despite a majority of these proteins functioning in ribosome pathway and protein processing, there may still be some proteins that are indispensable for the obligate intracellular parasitic life of EHP. Among these 479 proteins, there were 95 predicted transporters and 14 predicted secreted proteins (Appendix A). Among the 95 transporters, 91 were predicted by blast against Transporter Classification Database, and 4 were predicted by only gene or protein annotation. Moreover, 6, 2, and 1 were annotated as ATP, amino acid, and glucose transmembrane transporters, respectively. The existence of these transporters associated with energy and substrates transmembrane transport compensated for the metabolic deficiencies of EHP. As described above, EHP infection led to the down-regulation of hepatopancreas lipid and amino acid metabolism. Though EHP is deficient in de novo biosynthesis and degradation of fatty acid, long-chain fatty acid-CoA ligase indicated the processes associated with fatty acid biosynthesis and degradation happened in EHP cells.

## 3. Discussion

The hepatopancreas is an important multifunctional organ of Decapoda, acting as the largest organ of the digestive tract, the main organ of metabolism, and the major source of immune molecules. It is a paired, tubular gland, and the epithelial cells of the tubules are specialized for the digestion of food particles, nutrient uptake, glycogen, and lipid storage, generating molecules involved in innate immune response, synthesis and elimination of hormones, and so on [20,21,22]. There is no doubt that the nutrient absorption, metabolism, and immune defense of this organ would be influenced once challenged by a pathogen. Due to the intracellular parasitic features, the hepatopancreas, with its function in energy storage and metabolism, became the first choice of EHP. However, a large number of pathogens occupying the intracellular space will undoubtedly affect the function of the cells.

### 3.1. EHP Infection Resulted in Unbalance of Lipid Absorption, Metabolism, and Mediated Cell Signaling Process

Lipids are the main components of the hepatopancreas because of their role as the membrane structural building blocks and energy storage substance repositories [23,24]. Besides de novo synthesis, the hepatopancreas also reserves lipids by uptake from feeds, which requires a mass of mitochondria and sERs [20]. Microsporidia steal ATP from host cells to obligate intracellular parasitism by transporting them from cytosol or directly binding them to the mitochondria [25,26]. As a result, lipid absorption and catabolism, which need energy or mitochondria, are damaged and finally exhibit a decline in lipid storage and metabolism. In order to maintain intracellular life, EHP also must steal lipids for their membrane system building because of their deficient fatty acid biosynthesis [13,23]. Therefore, there is no difficulty in understanding why the fatty acid elongation and degradation, peroxisome, the biosynthesis of unsaturated fatty acid, as well as sphingolipid metabolism were down-regulated after EHP infection, nor why this regulation deteriorated with the aggravation of the infection. Down-regulated substance metabolism may be a strategy of the host to expand its lifespan in response to the “starvation” caused by infection.

Lipids like high-density lipoprotein (HDL), linoleic acids, and linoleic acid have been reported to be associated with non-specific immune responses [22,27,28]. Moreover, immune molecules such as lectins, hemocyanin, ferritin, antibacterial and antiviral proteins, proteolytic enzymes, and nitric oxide are mostly generated in hepatopancreas [21]. Our study showed the up-expression of lectin (partial isoforms were down-expressed), hemocyanin, and antimicrobial and antiviral proteins in the EHP-infected hepatopancreas, and the down-regulation of peritrophic and linolenic acid metabolism, which was related to the aggravation of infection. Considering this, we deem that an EHP invasion would activate the innate immune defense system (including lipid with immune defense function) of hepatopancreas, further influencing the metabolism of lipid and other materials. Unbalanced lipid metabolism leads to changes in lipid-mediated cellular signaling pathways like Toll and IMD signaling pathways, which in turn affect the immune system. The infection causes dynamic interactions of the immune response and metabolic processes.

Some pathogens deliberately seek lipid-rich host niches or enhance the availability of lipids by manipulating the host. Intracellular pathogens have evolved sophisticated mechanisms to manipulate and tap into the lipid metabolism of their host cells [29]. Apicomplexa transport host ceramides and cholesterol to meet the sphingolipid and cholesterol requirements for optimal growth and replication [30,31,32]. *Trypanosoma cruzi* and *Leishmania chagasi* benefit from the lipid of phagolysosomes for intracellular survival [33,34]. Dengue virus infection induces lipophagy for required energy production [35]. Pathogens use lipids not only as food or structural building blocks but also as important pathogenesis factors that allow the pathogen to evade immune responses [29]. Many viruses use cholesterol and sphingolipid-rich lipid rafts to get into and/or out of target cells to escape immune response [36,37]. In a word, lipids have a vital role in the host-parasite interaction, such as roles in nutrients and energy competition, inflammation and immunity response, pathogen proliferation, and the progression of pathogenesis. According to our data, EHP infection mainly influenced fatty acid metabolism-related lipid nutrients and energy supply, membrane lipid construction and cell signaling, and mechanistic unknown linoleic acid-related immune processes.

### 3.2. Down-Regulation of Energy Metabolism May Result from Long-Term Parasitism of EHP

Glycolysis and oxidative phosphorylation are the fundamental pathways of ATP generation in eukaryotes. All microsporidia have lost the pathways for oxidative phosphorylation and tricarboxylic acid (TCA). Moreover, most genes related to glycolysis have been lost in EHP. Thus, EHP relies entirely on importing ATP and metabolites from the host. Many parasites have been reported to enhance the energy metabolism of host cells by recruiting host glycolytic enzymes, activating the HIF-1α pathway, and disturbing mitochondrial function [38,39,40,41,42]. Some microsporidia can cause depletion of host glycogen and rapid glucose uptake during infection [43,44]. However, our data showed a full decline in oxidative phosphorylation, glycolysis, citrate cycle, and fatty acid oxidation after EHP infection. This is consistent with previous reports [45,46,47]. Although the glycolysis/gluconeogenesis and citrate cycle pathways were not so distinctively changed compared to other pathways, the expression of certain enzymes were significantly down-regulated, such as that of fructose 1,6-biphosphate aldolase, glyceraldehyde-3-phosphate dehydrogenase, pyruvate dehydrogenase E1 component subunit alpha, pyruvate dehydrogenase E1 component subunit beta, cytosolic malate dehydrogenase, isocitrate dehydrogenase, pyruvate Carboxylase, and betaine aldehyde dehydrogenase (Appendix A). Because of the low capacity of blood glucose control [24,48], the cell’s primary energy source, glucose, and glycogen would be consumed by EHP to a low level during the fasting phase, which then leads to a decrease in TCA cycle processing and consumption of storage fatty acid. The significantly decreased triacylglycerol lipase suggests a decline in lipid digestion and storage. With the depletion of intracellular energy materials, host cells may reduce their metabolic level by reducing the amount of enzymes and down-regulating enzyme activity. Considering the absolute energy dependence of EHP, the entire down-regulation of host energy metabolism seems unfavorable for its growth and proliferation. This may be a hosting strategy for chronic infection as our pathogenic samples were kept under infection for more than 20 days.

### 3.3. The Relevance of EHP Infection to Shrimp Growth

Glyoxylate metabolism (glyoxylate cycle) is a pathway by which fatty acids are converted into sugars. Down-regulation of glyoxylate metabolism, fatty acid synthesis, fatty acid degradation, and unsaturated fatty acid biosynthesis suggested a decrease in intracellular lipid. Up-expressed chitin binding protein and down-expressed chitinase indicated a low conversion level of chitin. It is well known that the shrimp hepatopancreas is the main storage organ, mainly accumulating lipids and a lesser degree of glycogen, which supply the required energy and chitin for molting. EHP infection led to the consumption of energy and nutrients, thereby destroying the energy and nutrients storage for molting and growth. In addition, individual growth and development are regulated by hormones, which usually contain JH and ecdysone. Our results showed that the JH degradation enzymes, JHE-like carboxylesterase 1 and Juvenile hormone epoxide hydrolase 1, declined in the EHP-infected hepatopancreas, suggesting that the accumulation of the JH in shrimp is due to the EHP infection. Ecdysteroid-regulated-like protein is negatively regulated by the ecdysteroid [46,49]. The significant up-regulation of this enzyme indicated a decrease in ecdysone. A high abundance of JH and low concentration of ecdysone meant delayed ecdysis and individual development. In conclusion, the capture of nutrients (lipid and amino acid) and energy coupled with the regulation of development-related hormones are the main reasons for the EHP-induced slow growth syndrome of the host.

## 4. Materials and Methods

### 4.1. Sampling and EHP Detection

Shrimp were obtained from individual freshwater ponds in Zhuhai, Guangdong province, China. Healthy shrimp from the normal pond and infected shrimp from the symptomatic pond were all selected for similar weight (4~6 g) and size (8~10 cm). The Hepatopancreas (HP) were promptly removed from shrimps in situ and quickly frozen by liquid nitrogen, subsequently delivered to the lab in dry ice, and stored at −80 °C. HP samples were immersed in liquid nitrogen and isolated in the appropriate tissue size for EHP detection. The tissue blocks were ground within liquid nitrogen and divided into two for PCR assay and microscopy. The genomic DNA (gDNA) was extracted with a 2% CTAB (pH 8.0) reagent, and EHP load was detected by SYBR green-based RT-qPCR with primers specifically recognizing gene *EHPptp2*. The reaction system and amplification process of RT-qPCR were referred to as the specification of the SYBR green mix (11201ES03, YEASEN, Shanghai, China), and each system contained 50 ng of hpDNA. Every template was analyzed in triplicate. For microscopy, Fluorescent Brightener 28 (FB28), which can bind with the chitin layer of EHP spores, was used to mark EHP [50]. The hepatopancreas was divided into three groups, the control (EHP-free), light (*EHPptp2* < 10^3^ copies/50 ng hpDNA, L group), and heavy (*EHPptp2* > 10^4^ copies/50 ng hpDNA, H group) groups. Each group contained four biological replicates, and every replicate was a mixture of four hepatopancreases with comparable infection levels.

### 4.2. Protein Extraction and Enzymolysis

The samples stored at −80 °C were fully ground to powder within liquid nitrogen. Four times the sample volume of phenol extraction buffer (with an equal volume of SDC and 1% protease inhibitor) was added to each sample, and ultrasonic lysis was performed. Then samples were centrifuged at 1000× *g* at 4 °C for 10 min. The phenol phase was treated at 100 °C for 5 min, followed by precipitate overnight with 5 volumes of 0.1 M ammonium acetate/methanol. The protein precipitate was washed with methanol and propanone, in sequence, and finally resuspended with 1% SDC. Proteins were quantified by a BCA kit (P0010, Beyotime, Shanghai, China).

Equal quantities of protein samples were enzymatically hydrolyzed, and the volume of each sample was adjusted to coincide with the lysate. Trichloroacetic acid (TCA) was gently added to a final concentration of 20%, followed by vortex mixing and precipitation at 4 °C for 2 h. The precipitate was then washed with pre-cooled acetone 2–3 times after centrifugation at 4500× *g* for 5 min. A final concentration of 200 mM tetraethylammonium bromide (TEAB) was added to dissolve the dried precipitate, which was then homogenized by ultrasonic and treated with trypsin in a 1:50 ratio (protease: protein, m/m) overnight. Then, 5 mM dithiothreitol (DTT) was added to reduce proteins at 56 °C for 30 min. Proteins were incubated with 11 mM iodine acetamide (IAA) for 15 min at room temperature, and away from the light in the end.

### 4.3. Liquid Chromatography-Mass Spectrometry (LC-MS/MS) Analyses

The peptides were dissolved in mobile phase A (an aqueous solution containing 0.1% formic acid and 2% acetonitrile) and separated using a NanoElute ultra-high performance liquid system (Bruker Corporation, Rheinstetten, Germany). Then a liquid phase gradient with mobile phase B (0.1% formic acid and 100% acetonitrile solution) was performed as follows: 6–24% B during 0–70 min, 24–35% B during 70–84 min, 35–80% B during 84–87 min, 80%B during 87–90 min; the flow rate was maintained at 450 nL/min. The separated peptides were injected into a capillary ion source for ionization and analyzed by timsTOF Pro mass spectrometry (Bruker Corporation, Germany). The primary ions and secondary fragments of peptides were assayed using a high-resolution TOF under a voltage of 1.75 kV. The scanning range of secondary mass spectrometry was set to 100–1700. In contrast, the data acquisition adopted parallel cumulative serial fragmentation (PASEF) mode. After the primary data of mass spectrometry were collected, a secondary spectrum with the charge of primary ions in the range of 0–5 was collected in PASEF mode for 10 times. The dynamic exclusion time of tandem mass spectrometry scanning was set to 30 s to avoid the repeated scanning of primary ions.

### 4.4. Spectrum Data Analysis

In this study, the secondary mass spectrometry data were retrieved using Maxquant (V1.6.15.0), setting parameters as follows: the reference database was UniProt and NCBI (*Penaeus vannamei*, assembly ASM378908v1; *Enterocytozoon hepatopenaei*, ASM2307953v1); adding a reverse database to calculate the false positive rate (FDR) caused by random matching, adding a common contamination database to eliminate the influence of contaminated proteins in identification results; the digestion mode was set as Trypsin/P; the number of missing digestion site was set to 2; the minimum peptide length was set to 7 amino acid residues; the maximum modification number of the peptide was set to 5; the mass error tolerance of primary ions in the First search and Main Search was set at 20 ppm; and that of secondary fragment ions was 20 ppm as well. The alkylation of cysteine Carbamidomethyl (C) was set as fixed modification, while variable modification was set as oxidation of methionine and N-terminal acetylation. The FDR of protein and PSM identification was set to 1%, as well as that of spectrogram, peptide, and protein identification. All identified proteins must have contained at least one unique peptide. The data reported in this paper were deposited in the OMIX, China National Center for Bioinformation/Beijing Institute of Genomics, Chinese Academy of Sciences (https://ngdc.cncb.ac.cn/omix: accession no. OMIX001256; accessed on 16 June 2022) [51,52].

### 4.5. Protein Annotation Methods

In order to obtain a thorough understanding of the functional characteristics of proteins, the annotation was performed with multiple methods. The detailed annotation includes Gene Ontology (GO), Kyoto Encyclopedia of Genes and Genomes (KEGG) pathway, and Subcellular localization. Gene Ontology (GO) annotation proteome was derived from the UniProt-GOA database (http://www.ebi.ac.uk/GOA/; accessed on 12 August 2021). Identified protein ID was converted to UniProt ID and then mapped to GO ID. If proteins were not annotated in the UniProt-GOA database, their GO function would be annotated according to sequence alignment using the InterProScan. Then proteins were classified by Gene Ontology annotation based on three categories: biological process, cellular component, and molecular function. The KEGG database was used to annotate protein pathways. Proteins were firstly annotated to a KEGG description using KEGG online service tools KAAS and then mapped on the KEGG pathway based on the annotation by the KEGG mapper. As to subcellular localization prediction, Wolfpsort, an updated version of PSORT/PSORT II for the prediction of eukaryotic sequences, was used in this study.

### 4.6. Function Enrichment and the Cluster of Differentially Expressed Proteins (DEPs)

The ratio of the mean of the relative quantitative values of each protein in repeated samples of comparison groups was defined as Fold Change (FC). In order to determine the significance of the difference, a T-test was performed on the relative quantitative value of each protein in the comparison group samples, and the corresponding *p* value was used as the significance index. We defaulted *p* value < 0.05 and defined the FC > 1.5 as significant up-regulation and FC < 1 or FC < 1.5 as significant down-regulation.

To confirm whether differentially expressed proteins have significant enrichment trends in certain functional types, we conducted an enrichment analysis of DEPs in each comparison group of the GO classification and KEGG pathway. They were evaluated by the *p* value of the two-tailed Fisher’s Exact test, and a corrected *p*-value < 0.05 is considered significant. Hierarchical clustering based on *p* values obtained from enrichment assay was applied to further explore the functional correlation of DEPs between different comparison groups.

### 4.7. Mfuzz Analysis

The Mfuzz method was used for cluster analysis of protein expression with the three grouped samples. In this method, a new clustering algorithm of fuzzy C-means algorithm was adopted. Compared with hard clustering algorithms such as K-means, it can reduce the noise interference on clustering results to a certain extent, and this algorithm can effectively define the relationship between gene and cluster. In order to further understand the biological processes involved in proteins in each cluster, we conducted an enrichment analysis of the KEGG pathway in each cluster.

### 4.8. RT-qPCR Verifying the mRNA Level of DEPs

RNA from the remaining tissue samples used for omics analysis were extracted with Trizol for differentially expressed gene verification. The Total RNA was retrotranscribed to cDNA using the Hifair^®^ II 1st Strand cDNA Synthesis SuperMix (YEASEN, Shanghai, China). RT-qPCR was carried out by the same method as described in Section 2.1. The transcripts of genes associated with fatty acid metabolism, carbohydrate metabolism, hormone metabolism, and amino acid metabolism were relatively quantified. *Tubulin alpha-I* was used as the reference gene. The relative ratio was calculated by the software LightCycler96 (Roche, Basel, Switzerland).

### 4.9. Transporter and Secreted Protein Prediction

Transporters were predicted by Transporter Classification Database (TCDB » HOME). The identified proteins were passed to signal peptide and transmembrane prediction program through SignalP-5.0 and DeepTMHMM. Signal peptide-containing proteins with or without only one transmembrane domain were considered secreted proteins.

## 5. Conclusions

To obtain more detailed information about the relationship between EHP infection and shrimp growth retardation, we investigated the proteome of the hepatopancreas of *L. vannamei* with a low and high load of EHP. Our results revealed that EHP infection caused an unbalance in the host metabolism and immunity, mainly inducing the down-regulation of energy generation, lipid and amino acid metabolism, ecdysteroid abundance, and Juvenile Hormone (JH) degradation (Figure 8). These changes were positively correlated with the severity of EHP infection. The differential regulation between heavy- and light-infected shrimp indicated that the down-regulated fatty acid degradation, alpha-linolenic acid, and sphingolipid metabolism, and the up-regulated ubiquitin-mediated proteolysis and accumulation of JH may be the dominant metabolic reasons for the growth disorder caused by EHP. Thus, EHP infection-induced energic and structural lipid consumption and hormone regulation disorder are mainly responsible for the stunted growth.

## Figures and Tables

**Figure 1 ijms-23-11574-f001:**
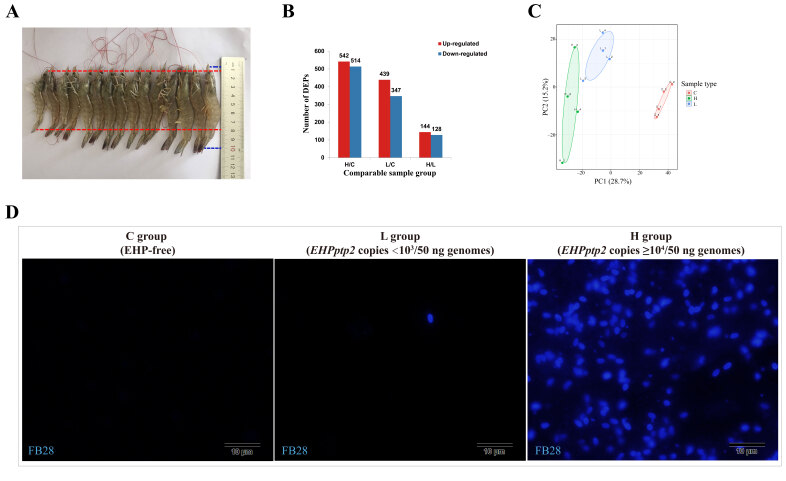
Sample information and basic statistics of proteomic data. (**A**): Shrimps in the pathogenetic pond presented various sizes. The large ones reached 10 cm (blue dotted line), while the little ones were less than 8 cm (red dotted line). (**B**): Summary of differentially expressed proteins, and the up- and down-regulated numbers were represented with red and blue columns, respectively. (**C**): PCA analysis was used to assess the repeatability of samples. (**D**): Shrimp samples were divided into three groups based on EHP load quantified by *EHPptp2* RT-qPCR. The EHP quantity of each group was more intuitive by fluorescence microscopy, as the fluorescent brightener 28 (FB28) could label the spore wall of EHP. Each blue, fluorescent spot represented an EHP spore.

**Figure 2 ijms-23-11574-f002:**
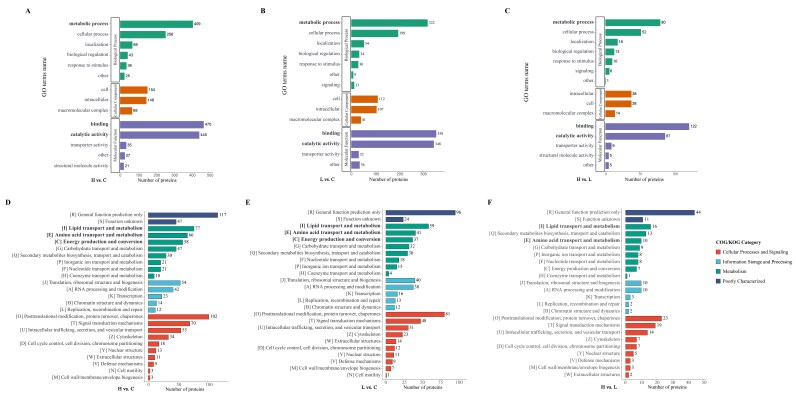
The function of DEPs classified by GO and GOC. (**A**–**C**): Go classification revealed that proteins involved in the cellular metabolism process with binding or/and catalytic activity functions were regulated after EHP infection (green and purple columns). The length of the column represents the enriched proteins. The comparison of the results also suggested that the quantitative change of host proteins was related to the EHP level. (**D**–**F**): COG classification indicated that the regulated proteins in EHP-infected hepatopancreas mainly play a role in lipid and amino acid transport and metabolism processes (green columns with bold).

**Figure 3 ijms-23-11574-f003:**
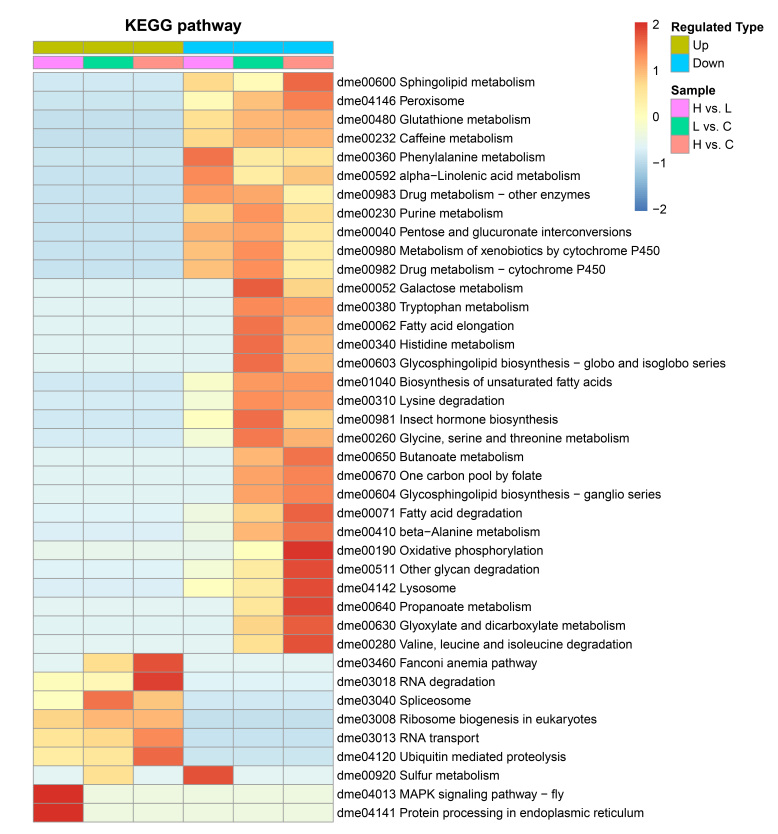
Functional enrichment clusters of differentially expressed proteins. KEGG enrichment of differentially expressed proteins. One color block represents a functional pathway of DEPs enrichment in different comparison groups, red represents a high degree of enrichment, and blue represents weak enrichment. The entire enriched KEGG pathways mainly manifested as down-regulated lipid and amino acid metabolism, and the regulation level between the H and L groups was discrepant.

**Figure 4 ijms-23-11574-f004:**
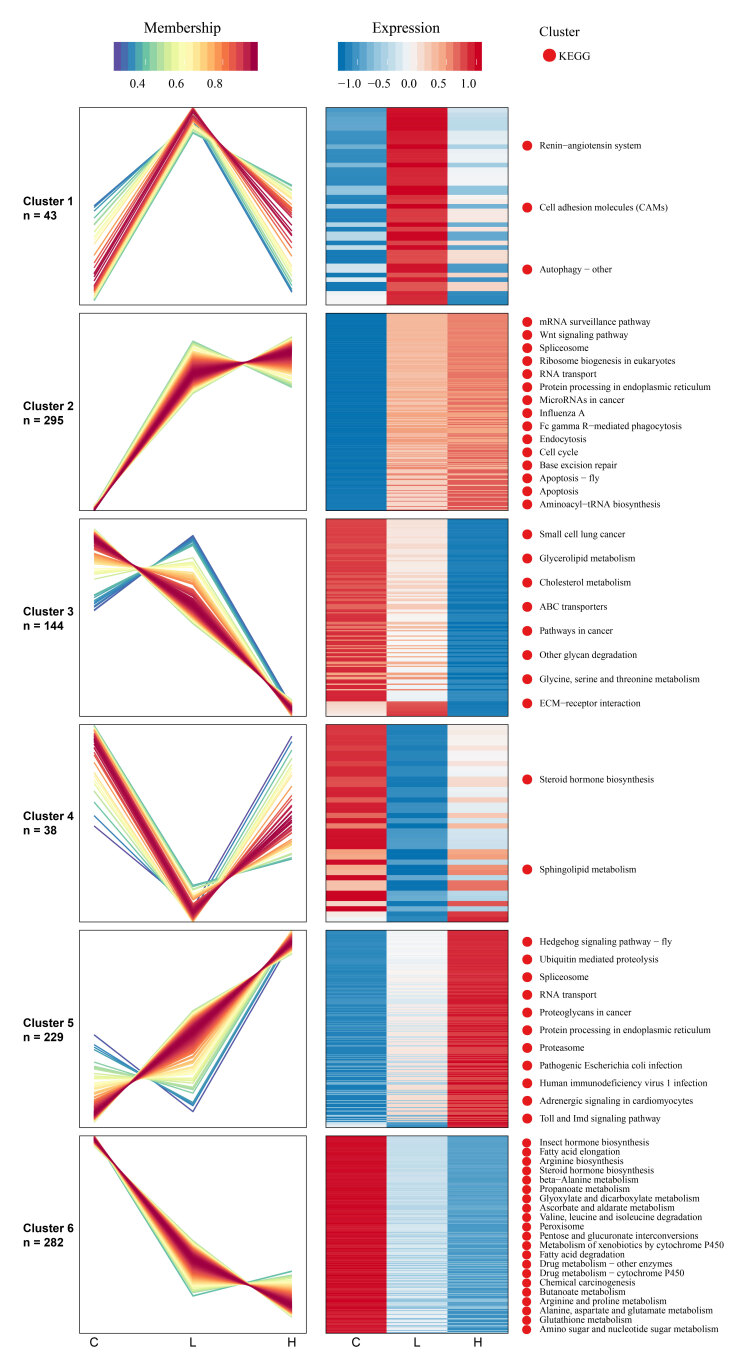
Expression pattern of identified proteins and modification sites via Mfuzz analysis. Mfuzz analysis result of the proteome. The left broken line charts show the protein expression trends in each sample. The horizontal axis indicates different groups, the vertical axis indicates the protein expression amount, and one broken line represents one protein. According to the different trends of the broken lines, they were divided into six different trends through cluster analysis. Heat maps were drawn for the protein sets in each cluster. To further understand the biological processes in which proteins in each cluster are involved, we carried out an enrichment analysis of the KEGG pathway of proteins in each cluster. The first few of the most significant enrichment were listed on the right.

**Figure 5 ijms-23-11574-f005:**
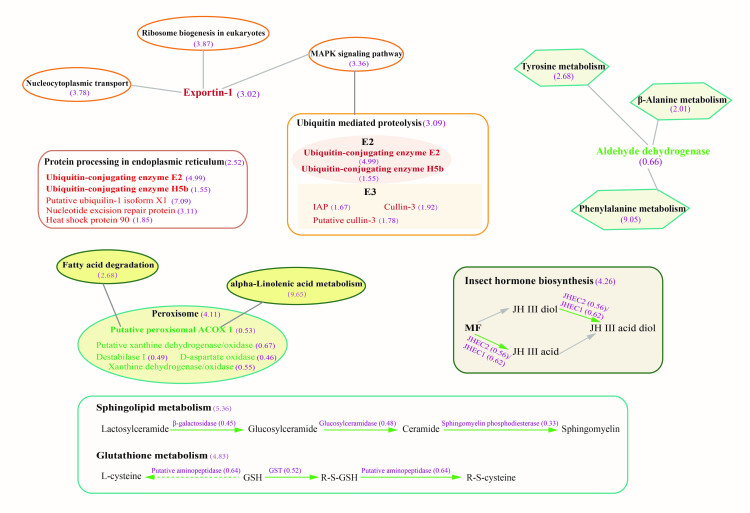
Summary of the main differentially regulated pathways between the H and L group. Green-colored ellipses, rectangular frames, and polygons indicate down-regulated pathways. In comparison, the orange-colored ones indicate up-regulated processing. Purple-colored ones indicate the participant enzymes or the fold enrichment value of metabolism pathways, and the value in brackets represents the ratio of H to L.

**Figure 6 ijms-23-11574-f006:**
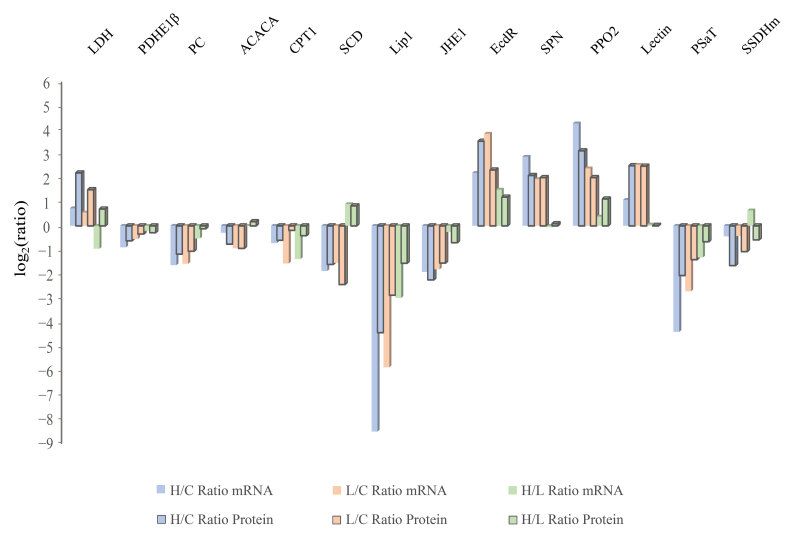
Transcription detection of target genes. Relative quantitation RT-qPCR verified the mRNA level of genes correlated with fatty acids, carbohydrates, hormones, and amino acid metabolism. The log_2_ (relative ratio) of target genes was used to assess the condition of gene regulation. Different colored columns represent different comparative levels. LDH: lactate dehydrogenase; PDHE1β: pyruvate dehydrogenase E1 component subunit beta; PC: pyruvate carboxylase; ACACA: acetyl-CoA carboxylase; Lip1: Triacylglycerol lipase; CPT1: carnitine O-palmitoyltransferase 1; SCD: Acyl-CoA delta-9 desaturase; JHE1: JHE-like carboxylesterase 1; EcdR: Ecdysteroid regulated-like protein; SPN: pacifastin light chain-like serine proteinase inhibitor; PPO2: Prophenoloxidase-2; PSaT: phosphoserine aminotransferase; SSDHm: mitochondrial succinate-semialdehyde dehydrogenase.

**Figure 7 ijms-23-11574-f007:**
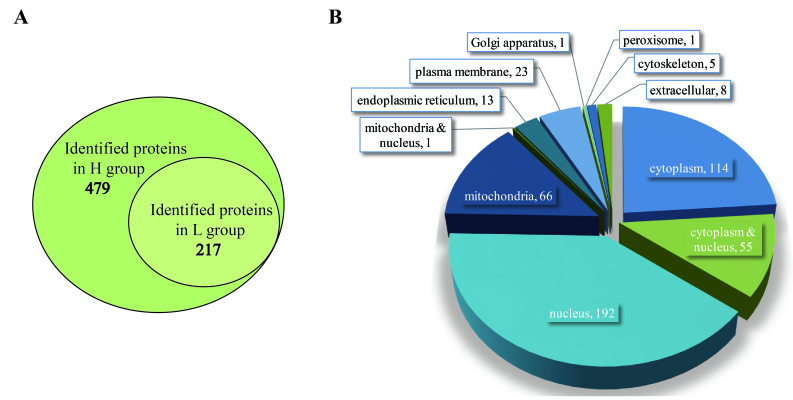
Information of identified pathogen proteins in the H and L group. (**A**): Statistic data of EHP proteins. The outer ellipse represents 479 EHP proteins found in the H group, and the inner ellipse indicates 217 EHP proteins were found in the L group. (**B**): Predicted subcellular location of EHP proteins found in infected groups. Protein numbers were shown on or side of each item.

**Figure 8 ijms-23-11574-f008:**
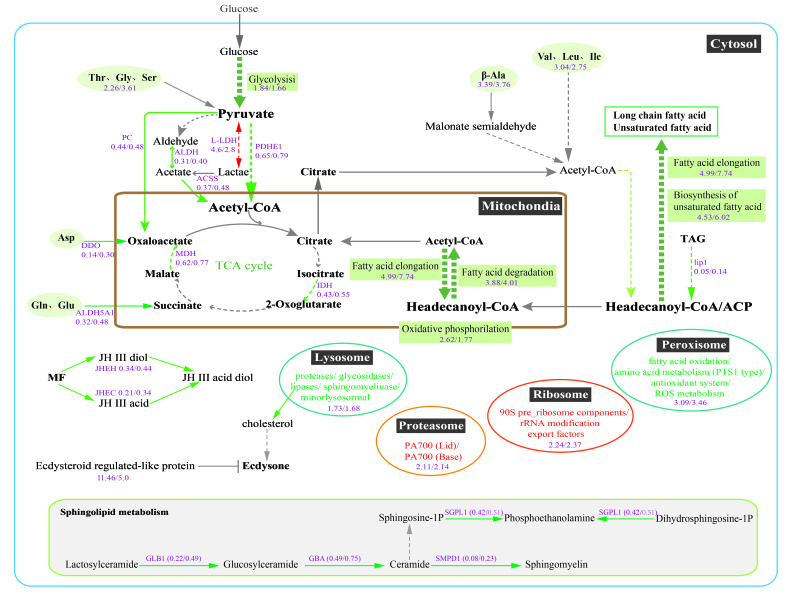
Overview of the main regulatory pathways in the EHP-infected hepatopancreas. The green color indicates pathways, and the arrows represent down-regulated metabolic processing and reaction steps. The red-colored arrow represents up-regulated reaction steps and orange colored indicates up-regulated processing or proteins. The purple color indicates the participant enzymes or the fold enrichment value of metabolic pathways, and the value in front of the diagonal represents the ratio of H vs. C, while the value behind the diagonal represents the ratio of L vs. C.

## Data Availability

The data reported in this paper have been deposited in the OMIX, China National Center for Bioinformation/Beijing Institute of Genomics, Chinese Academy of Sciences (https://ngdc.cncb.ac.cn/omix: accession no. OMIX001256; accessed on 16 June 2022).

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
