# Peer review of "Down-Regulation of Lipid Metabolism in the Hepatopancreas of Shrimp Litopenaeus vannamei upon Light and Heavy Infection of Enterocytozoon hepatopenaei: A Comparative Proteomic Study"

_ijms, 2022, doi:10.3390/ijms231911574_

Round 1

Reviewer 1 Report

Can you please add in the material and methods section, how many hepatopancreas were sampled? Were they sampled in situ? Also, it is confusing at it is written, how many hepatopancreas samples were considered in each experimental group (control), low and high EHP, please include this information.

It is known in crustaceans that the steadily maturing character of the mounts is affected by steadily decreasing levels of juvenile hormone. However, line 350 you described the opposite, please verify it. 

Author Response

We appreciate you for spending time to review this article. You have given us professional advices and improved the quality of the article. We sincerely thank you for that. We are very sorry for the linguistic problem in the previous version. The language of revised manuscript has been corrected and modified by a qualified person.

The following are our responcse to your comments.

Point 1: Can you please add in the material and methods section, how many hepatopancreas were sampled? Were they sampled in situ? Also, it is confusing at it is written, how many hepatopancreas samples were considered in each experimental group (control), low and high EHP, please include this information.

Response 1: The analyzed hepatopancreas were directly sampled in situ with properly handled and preserved in low temperature. As we described at line 361-363 that “Hepatopancreas(HP) were promptly removed from shrimps and quickly frozen by liquid nitrogen, subsequently delivered to the lab in dry-ice and stored at -80℃.” In addition, each group consisted of four biological replicates and every replicate was a mixture of four hepatopancreas with comparable infection level. It means that each experimental or control group contained 16 hepatopancreas samples. We have discribed in line 89-90 and also added these information to methods section (line 374-376).

Point 2: It is known in crustaceans that the steadily maturing character of the mounts is affected by steadily decreasing levels of juvenile hormone. However, line 350 you described the opposite, please verify it.

Response 2: We accept the opinion that the maturity of crustaceans is affected by the decrease of juvenile hormone. It is probable that our language description mislead you. The description in line 353-354 “A high abundance of JH and low concentration of ecdysone mean delayed ecdysis and individual development.” also intended to express that the inhibition of juvenile hormone degradation blocked the maturation of the host.

Reviewer 2 Report

Abstract: Ok

Introduction: OK

Figure 1 A, the first picture is needless

Figure 1 C, meaning that no protein?? make clear or may be problem about the figure resolution

Fig 4 very nice designed

check whole english of manuscript

Author Response

We appreciate you for spending time to review this article. You have given us professional advices and improved the quality of the article. We sincerely thank you for that.

The following are our responcse to your comments.

Point 1: Figure 1 A, the first picture is needless.

Response 1: We accept your suggestion and deleted the picture. The existence of the figure 1A was just to let audience have an visualized view of sampling environment, and deleting it would not affect the understanding of the article.

Point 2: Figure 1 C, meaning that no protein? make clear or may be problem about the figure resolution.

Response 2: The fluorescent brightener 28 (FB28) is a fluorescent dye that can specifically bind to the β-1, 4 glycosidic bonds of chitin, which is one of the main components of microsporidian spore wall. Therefore, FB28 is widely used to stain microsporidian spores. The pathogen load in the hepatopancreas can be more visually understand from fluorescently labeled microsporidia spores. More detailed description had been added in the revised manuscript as follows.

Line 107-109: The EHP quantity of each group was more intuitive by fluorescence microscopy, as the fluorescent brightener 28 (FB28) could label the spore wall of EHP. Each blue fluorescent spot represented an EHP spore.

Line 370-372: For microscopy, Fluorescent Brightener 28 (FB28), which can bind with the chitin layer of EHP spores, was used to label EHP spores.

Point 3: Fig 4 very nice designed.

Response 3: Thanks for your affirmation.

Point 4: check whole english of manuscript

Response 4: We are very sorry for the linguistic problem in the previous version. The language of revised manuscript has been corrected and modified by a qualified person.